# Unconventional charge order in a co-doped high-$T_c$ superconductor

D. Pelc[1], M. Vučković[1], H.-J. Grafe[2], S.-H. Baek[2] & M. Požek[1]

Charge-stripe order has recently been established as an important aspect of cuprate high-$T_c$ superconductors. However, owing to the complex interplay between competing phases and the influence of disorder, it is unclear how it emerges from the parent high-temperature state. Here we report on the discovery of an unconventional ordered phase between charge-stripe order and (pseudogapped) metal in the cuprate $La_{1.8-x}Eu_{0.2}Sr_xCuO_4$. We use three complementary experiments—nuclear quadrupole resonance, nonlinear conductivity and specific heat—to demonstrate that the order appears through a sharp phase transition and exists in a dome-shaped region of the phase diagram. Our results imply that the new phase is a state, which preserves translational symmetry: a charge nematic. We thus resolve the process of charge-stripe development in cuprates, show that this nematic phase is distinct from high-temperature pseudogap and establish a link with other strongly correlated electronic materials with prominent nematic order.

[1] Faculty of Science, Department of Physics, University of Zagreb, Bijenička 32, Zagreb HR 10000, Croatia. [2] IFW Dresden, Institute for Solid State Research, P.O. Box 270116, Dresden D-01171, Germany. Correspondence and requests for materials should be addressed to D.P. (email: dpelc@phy.hr) or to M.P. (email: mpozek@phy.hr).

Cuprate high-temperature superconductors display a staggering complexity of electronic behaviour, which is the subject of intense research and controversy. One of the central questions of cuprate physics is the role of competing electronic ordering tendencies in determining both normal-state and superconducting properties. Recently, charge-stripe order has emerged as a ubiquitous phenomenon in cuprates[1–7] and charge stripes have been proposed to play a role in almost all of the important features of the cuprate phase diagram: the mysterious pseudogap phase[8,9], Fermi surface reconstruction[10] and even superconducting pairing[11]. However, firm experimental evidence for these claims is difficult to obtain, because charge stripes are sensitive to disorder, which is a prominent feature of all cuprate materials[12]. Thus, true long-range charge-stripe order is never established and a complex glassy/fluctuating stripe dynamics emerges[9,13,14]. Adding to the intrigue, several theoretical studies predict that charge stripes appear through unconventional precursor phases[15–18], such as a charge nematic, which do not break translational symmetry at all. Although electron nematic ordering has been discussed extensively in other strongly correlated materials such as pnictide superconductors[19,20] and quantum Hall systems[21], its existence in the cuprates is less clear[22–26]. Thus, gaining insight into the physics of the emergence of charge stripes from a parent high-temperature metallic state becomes of great interest.

Here we combine an unusual experimental technique— nonlinear conductivity—with two established ones—nuclear quadrupole resonance (NQR) and specific heat—to study the appearance of charge stripes in the cuprate $La_{2−x−y}Eu_ySr_xCuO_4$ with $y = 0.2$ (LESCO). We use LESCO as a model system because of its fortunate arrangement of structural and charge-related transitions. It is well known that lanthanum cuprates in their low-temperature tetragonal (LTT) phases have prominent charge and spin stripe order; the archetypal example is $La_{2−x}Ba_xCuO_4$ (LBCO), where superconductivity is strongly suppressed by stripe order around $x = 1/8$, and separate charge and spin stripe ordering temperatures are observed in neutron scattering, transport and magnetic susceptibility[27–29]. However, in LBCO the static stripes only exist in the LTT phase and abruptly disappear on heating above the tetragonal-to-orthorhombic (LTT/LTO) transition, precluding any investigation of their intrinsic melting mechanism in the tetragonal setting. We therefore use the europium co-doped LESCO as a representative system for studying charge-stripe physics, owing to static charge stripes disappearing spontaneously far below the LTT/LTO transition temperature[5,30–32].

Our work provides unprecedented experimental insight into the dynamics and thermodynamics of the process of charge-stripe formation in the cuprates. We show that indeed the stripes develop through an unconventional precursor phase consistent with a charge nematic. This phase is found to be insensitive to disorder, appearing at a true thermodynamic phase transition, in agreement with recent theoretical predictions for nematics[9]. In contrast, the transition into the charge-stripe phase is smeared out by disorder, resulting in glassiness and short-range stripe correlations. Importantly, we find that the nematic order closely follows the dome-shaped charge-stripe appearance in the phase diagram, implying that it is separate from the high-temperature pseudogap in LESCO.

## Results

**Nuclear magnetic resonance**. NQR experiments on copper, specifically measurements of NQR signal intensity in dependence on temperature, provide the first indication that an additional phase exists between charge order and the metallic state in LESCO. A decrease of Cu NQR signal—termed the wipeout effect—has been previously observed in charge-ordered phases of several cuprates, including LBCO and LESCO[13,33–37]. In our experiments, strong wipeout also begins at the charge-order onset temperatures $T_{CO}$ obtained by resonant X-ray scattering[31]—but in addition a wipeout plateau is observed, extending 10–20 K above $T_{CO}$ and ending at a temperature $T_{EN}$, significantly below the structural transition (Fig. 1). For doping $x = 0.125$, we have performed an additional Cu nuclear magnetic resonance (NMR) measurement in a field of 11T, perpendicular to the crystalline $c$ axis to obtain pure exponential spin–spin relaxation[38]. Owing to a measuring frequency $\sim 4$ times higher than in NQR, the NMR wipeout results have significantly better signal-to-noise ratios and the measurement confirms the existence of an intermediate phase (Fig. 1). Interestingly, the wipeout fraction in 11T is systematically lower than in pure NQR by a factor of $\sim 1.6$, implying that the strong in-plane field modifies the fluctuations responsible for wipeout. A similar influence is observed[39] in La NMR of LBCO-1/8. The detailed investigation of magnetic field effects is left for future discussion.

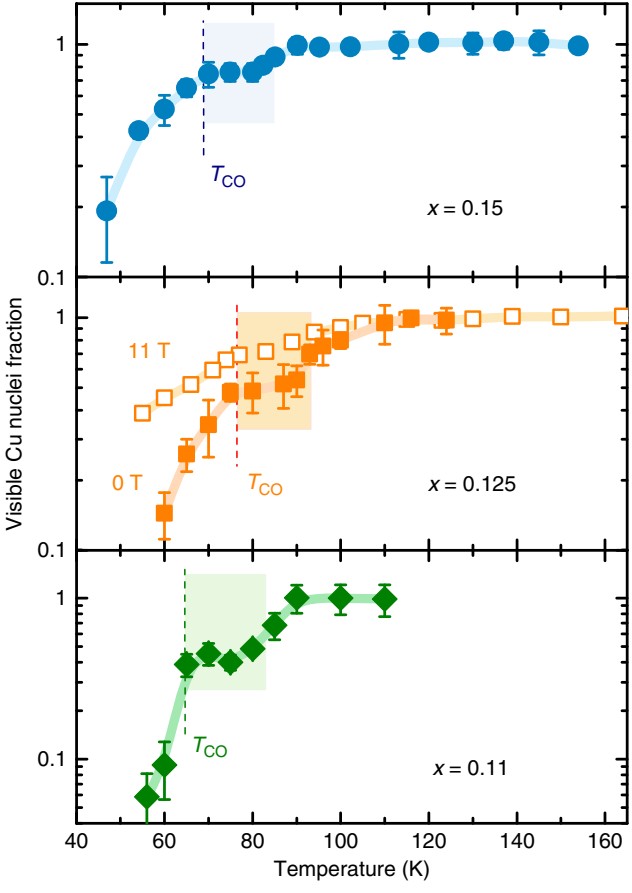

**Figure 1 | Copper NQR signal wipeout.** NQR measurements of $^{63}Cu$ signal intensity in three $La_{1.8−x}Eu_{0.2}Sr_xCuO_4$ samples, compensated for spin–spin relaxation and Boltzmann temperature dependence, and normalized to high-temperature values. Lines are guides to the eye and error bars are s.d. of the spin–spin decay exponential least-square fits. The intermediate phase above the charge-ordering temperature is clearly seen as a plateau in the wipeout fraction; the logarithmic scale is noteworthy. Empty squares for $x = 0.125$ are an additional Cu NMR measurement in an external field of 11T, also displaying a wipeout plateau. The structural orthorhombic-tetragonal (LTO-LTT) transition is at $\sim 130$ K for all samples and the charge-stripe ordering temperatures $T_{CO}$ are from a resonant X-ray study[31].

Although Cu wipeout has been studied before in LESCO and related compounds, to our knowledge the plateau we see here has not been observed before. There are several possible reasons for this: the use of powders instead of single crystals, insufficient resolution and the dependence of wipeout on magnetic field, which has not been systematically investigated. Here, both temperature and wipeout resolution is significantly higher than in previous studies, we only work with single crystals to eliminate powder sample/grain boundary issues[13,33,36,37] and we have carefully avoided or compensated for spurious effects, which could modify the signal intensity. Structural changes cannot influence the results (the LTO/LTT transition is close to 130 K in the entire investigated doping range[31,40] and no significant intensity change is seen at $T_{LTT}$), spin–spin relaxation is compensated, for using exponential fits (see Supplementary Figs 1 and 2), the NQR lineshape remains essentially the same throughout our temperature range[13,34] (see Supplementary Fig. 3) and skin depth changes at MHz frequencies are of the order of 1% in our range of temperatures, too small to account for the signal change (see Supplementary Fig. 4 and Methods for more details on these effects and procedures). However, we stress that the microscopic mechanism causing wipeout is a matter of debate and without a deeper understanding of its origin, wipeout remains a blunt instrument for studying electronic order. Here, its main purpose is to show that two distinct characteristic temperatures are always present, allowing us to construct a phase diagram.

A close relationship between stripe order and the intermediate phase can be inferred from the behaviour of transition temperatures at different doping concentrations. We define the intermediate phase transition temperature $T_{EN}$ as the midpoint of the wipeout step (the values agree with specific heat results, to be discussed below). $T_{EN}$ approximately follows the dependence of $T_{CO}$ on Sr concentration (Fig. 2). Another appealing feature of the wipeout plateau is the dependence of its height (compared with high-temperature values) on doping (Fig. 2, empty squares), showing a trend different from the transition temperatures. It appears that the intermediate phase either merges with the stripe phase or vanishes altogether at $x \approx 0.17$. The LTT structure becomes unstable at similar dopings[31,40], although previous measurements do give indications of stripe order persisting up to $x \sim 0.2$ (refs 33,36).

As a local probe, NQR cannot tell us more about the long-range properties of the intermediate phase or whether it is a distinct state of electronic matter at all. As there is no sign of the intermediate phase in X-ray scattering—which needs broken translational symmetry and coherence lengths of at least a few nanometres—we conclude that the order is either short ranged or possesses higher symmetry than charge stripes. This leads to two possible scenarios: phase separation, wherein nanoscale stripe-ordered bubbles are mixed with regions of pseudogapped metal, or a lower symmetry phase (stripe liquid), with restored long-range translational symmetry, but rotational symmetry still broken. To decide between the two possibilities we perform two additional experiments on the sample with $x = 1/8$—nonlinear conductivity and heat capacity measurements.

**Nonlinear conductivity.** Nonlinear conductivity is generally understood as the leading order correction to conventional Ohm's law:

$$j = \sigma E + \sigma_3 E^3 + \dots \qquad (1)$$

where $j$ is the current density, $\sigma$ the linear conductivity and $\sigma_3$ the leading nonlinear correction—if time-reversal or inversion symmetry is not broken, only odd powers of $E$ can enter relation (1).

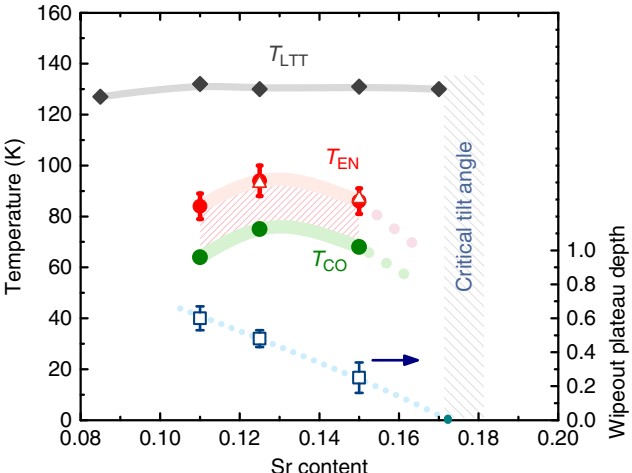

**Figure 2 | Phase diagram of La$_{1.8-x}$Eu$_{0.2}$Sr$_x$CuO$_4$.** The intermediate phase, as obtained from NQR wipeout and specific heat, approximately follows the charge-stripe onset temperatures $T_{CO}$ (green circles, from resonant X-ray; ref. 31) for the samples investigated. The structural transition temperatures $T_{LTT}$ (grey diamonds) are from ref. 40 and significantly higher than either $T_{CO}$ or $T_{EN}$. The transition temperatures $T_{EN}$ (red circles) are determined as the midpoint of the wipeout steps, with the error bars denoting the widths of the steps. These $T_{EN}$ from NQR agree with transition temperatures seen in specific heat, for two samples (empty triangles). The height of the wipeout plateau (empty squares) has a different trend with increasing Sr content, indicating that the intermediate phase disappears at a critical doping slightly above $x = 0.17$, which approximately coincides with the critical tilt angle of Cu–O octahedra beyond which the tetragonal (LTT) phase is unstable. Error bars are s.d. from the means of the wipeout steps.

Relation (1) is written in a schematic way, as $\sigma_3$ is in general a fourth rank tensor. In our experiment the sample is oriented so that the electric fields and nonlinear currents lie in the CuO$_2$ planes; thus, we only detect planar components of the $\sigma_3$ tensor (see Supplementary Note 1 for details). Owing to high linear conductivity of doped cuprates, measuring $\sigma_3$ without significant unwanted heating effects is challenging. We have therefore developed a contactless pulse method[41], wherein currents are induced in the sample at some frequency $\omega$ (typically $\sim 20$ MHz) and the response at $3\omega$ (which is proportional to $\sigma_3$) is detected. Results of such a measurement are shown in Fig. 3, with two characteristic features clearly resolved: charge order below $\sim 75$ K, where a nonlinear signal is expected due to either stripe pinning dynamics[42,43] or glassiness[44,45], and the dramatic peak at $T_{EN}$. We stress that the linear conductivity is smooth and almost featureless (see Supplementary Fig. 4). Yet, small deviations from high-temperature behaviour were previously detected close to $T_{EN}$ even in linear transport measurements on LESCO[10,46]— possibly another consequence of the unconventional order.

**Specific heat.** The peak in $\sigma_3$ is highly suggestive of a diverging susceptibility at a phase transition. If a phase transition indeed occurs at $T_{EN}$, it must have a signature in the specific heat, regardless of its microscopic nature. We succeeded in detecting such a signature in a sensitive differential calorimetry experiment. The results for the sample with $x = 0.125$ are shown in Fig. 4, with two visible features—the structural transition at $T_{LTT} \sim 130$ K and a distinct peak at $T_{EN}$. A comparison of nonlinear conductivity and $\Delta C_p$ (inset) displays the concordance between transition temperatures. It is noteworthy that the $\Delta C_p$ in the inset of Fig. 4 is a different, higher resolution measurement than the one in the

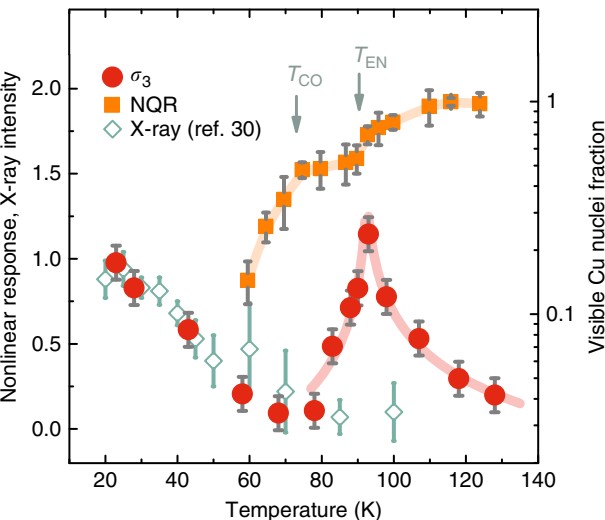

**Figure 3 | Third harmonic response in La$_{1.675}$Eu$_{0.2}$Sr$_{0.125}$CuO$_4$.** The nonlinear conductivity $\sigma_3$ is shown in dependence on temperature and normalized to the low-temperature value (circles), with the electric fields in the CuO$_2$ planes. Error bars are s.d. from measurements at different electric fields (see Methods). A sharp peak is observed at the intermediate phase transition temperature $T_{EN}$, in good agreement with nucelar quadrupole resonance wipeout measurements (squares; error bars are s.d. of the spin–spin decay exponential least-square fits). The linear conductivity is featureless in the entire temperature range. Solid lines are a guide to the eye. The nonlinear signal closely follows the charge-stripe peak intensity (diamonds) obtained in a resonant X-ray study[30] below $T_{CO}$, but not at higher temperatures. This implies that the unconventional phase between $T_{CO}$ and $T_{EN}$ is qualitatively different from charge stripes and does not break translational symmetry.

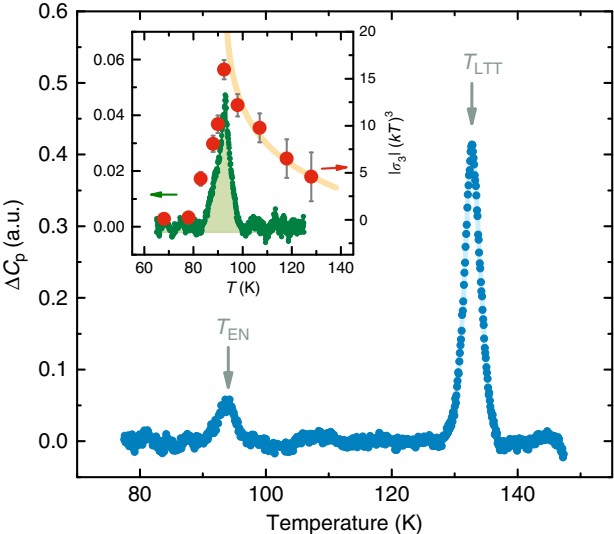

**Figure 4 | Differential heat capacity measurement in La$_{1.675}$Eu$_{0.2}$Sr$_{0.125}$CuO$_4$.** Two features can be discerned—the orthorhombic-tetragonal structural transition at $T_{LTT} \sim 130$ K and a signature of the intermediate phase at $T_{EN}$. The latter is strong evidence that a true thermodynamic phase transition takes place close to $T_{EN}$. Inset shows a comparison between the quadrupolar correlation function $|\sigma_3|(kT)^3$ (circles; error bars are s.d. from measurements at different electric fields) and $\Delta C_p$ (small circles) for temperatures in the vicinity of $T_{EN}$. The $\Delta C_p$ measurement in the inset is performed $\sim 10$ times slower than in the main graph, enhancing the sharpness of the cusp at $T_{EN}$. The solid line is a fit to a logarithmic divergence $\ln |(T - T_{EN})/T_{EN}|$, showing that no simple power law (scaling) relation can describe the data.

main graph, demonstrating both the repeatability and sharpness of the peak at $T_{EN}$. An additional measurement was performed on the sample with $x = 0.15$, confirming the presence of the transition (see Supplementary Fig. 5). The transition temperatures $T_{EN}$ from all three methods—NQR, nonlinear conductivity and specific heat—agree very well, justifying our definition of $T_{EN}$ as the midpoint of the wipeout step in Fig. 2. Notably, no sharp features are present at $T_{CO} \sim 75$ K: this is to be expected, as charge stripes are sensitive to disorder and only in a clean material would they occur through a true phase transition.

## Discussion

Our combined experimental approach establishes the existence of a well-defined phase between charge-stripe order and the metallic state in LESCO. Furthermore, nonlinear response and specific heat strongly suggest that a distinct type of order exists in the intermediate phase. Namely, the ordering is insensitive to structural disorder present in all doped cuprates, in contrast to stripes: orthorhombic twin domains create pinning centres for stripe order, making it glassy and smearing out the stripe ordering transition[2,9,13]. The temperature $T_{CO}$ is thus not a true phase transition, but rather an onset temperature where translational-symmetry breaking stripe correlations first appear. The intermediate ordering transition, on the contrary, remains thermodynamically well defined. Thus, the intermediate phase order clearly behaves in a qualitatively different way from charge stripes, either homogeneous or phase separated. Although the nonlinear signal close to $T_{EN}$ might also be a dynamical, glassy effect[44], we believe that this is not the case, for several reasons as follows: the transition is sharp, which would be odd for a glass transition; within experimental sensitivity, we have not observed

hysteresis or ageing effects characteristic of glassy systems (including spin glasses, which do occur in cuprates at lower temperatures); and the peak-like signature in heat capacity can only mean a true phase transition.

Having established the existence of the intermediate ordering phase, we now show it is consistent with a charge nematic. A recent theory[18] predicts a series of phase transitions between charge-stripe order and Fermi liquid (in a clean material), in line with our observations. In that scenario, the intermediate phase would correspond to a nematic stripe loop metal—a state with broken rotational symmetry, essentially a stripe liquid with proliferated double defects (stripe loops), with a preferred direction (see schematic depiction on Fig. 5). Related nematicity has been extensively discussed in strongly correlated systems[5,9,15,16,19,20,22,23]. In addition, a recent field-theoretical investigation has demonstrated that even weak quenched disorder destroys long-range charge-stripe correlations—precluding a sharp transition in the charge-stripe phase—whereas the charge nematic transition remains well-defined[9], in agreement with our data. A charge nematic has been shown to induce spin stripe fluctuations if the system is close to an antiferromagnetic instability[47], which is the case in cuprates. Such strong local spin fluctuations provide a natural explanation of the observed wipeout plateau[34,35]. Normally, the most direct sign of rotational symmetry breaking is the resulting conductivity anisotropy, as for example, in quantum Hall systems[21], but in the tetragonal phase of LESCO the nematic ordering direction is expected to vary from one CuO$_2$ plane to the next[28], so that no macroscopic anisotropy appears in linear conductivity. However, under quite general conditions, an extended fluctuation-dissipation relation connects the quantity $|\sigma_3|(kT)^3$ to four-point, quadrupolar charge correlations[45,48], which have the same symmetry as nematic fluctuations (see Supplementary Note 1). In other words, the

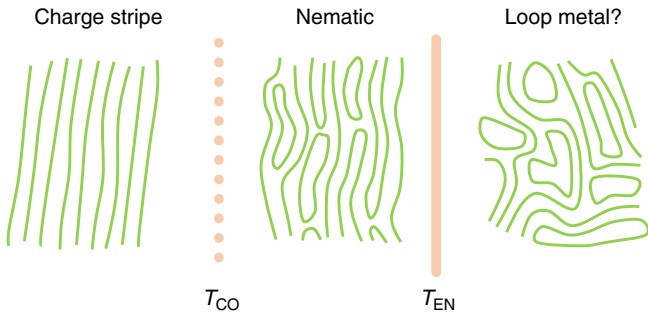

**Figure 5 | Electronic liquid crystal physics in La$_{1.8-x}$Eu$_{0.2}$Sr$_x$CuO$_4$.** Three proposed phases are depicted schematically, according to the stripe melting model of refs 17,18: the charge-stripe phase below $T_{CO}$, the loop nematic phase between $T_{CO}$ and $T_{EN}$, and the (tentative) loop metal phase with strong nematic fluctuations above $T_{EN}$. Only joined stripe ends exist in both nematic and loop metal phase; other stripe lines are assumed to extend out of the pictures. In a realistic, disordered material, the phase transition at $T_{CO}$ is smeared out and charge stripes become glassy, whereas the nematic order remains sharply defined.

connection between third harmonic response and nematic fluctuations originates from the quadrupolar nature of nematic order[16,45,48,49]. All two-point correlations, and thus linear response components, are featureless, because in a nematic only the direction (and not the relative sign) of electronic density modulations is important. Consequently, nematic order couples to other probes sensitive to quadrupolar effects, such as Raman scattering[50] and NQR spectroscopy[51]. Yet, the latter connection is not useful in La-based cuprates due to the great width of the Cu NQR lines and strong antiferromagnetic fluctuations, only leaving us with the possibility to observe the indirect influence of the nematic on NQR wipeout. The above symmetry-based reasoning is rather general: higher-order correlators also appear in spin glass/spin nematic physics, where translational symmetry remains unbroken as well. Therefore, $|\sigma_3|(kT)^3$ is a measure of charge nematic fluctuations and it diverges at the transition temperature $T_{EN}$, strongly suggesting that the transition is into a nematic phase.

Curiously, the temperature dependence of $|\sigma_3|(kT)^3$ does not follow any scaling relation (although the small range below $T_{EN}$ precludes any strong conclusions there). Instead, it seems to possess a logarithmic divergence of the form $|\sigma_3|(kT)^3 \sim \ln|(T-T_{EN})/T_{EN}|$ (inset of Fig. 4). Although it is difficult to reliably differentiate between a logarithmic dependence and possible corrections to scaling, either way we have an indication that any mean-field description of fluctuations above $T_{EN}$ breaks down. It is tempting to speculate that this is related to the most intriguing prediction of the theory of stripe melting[17,18]—the existence of another exotic metallic phase between charge nematic and Fermi liquid, a stripe loop metal, which should exhibit strong quadrupolar fluctuations[18] and thus large $\sigma_3$.

The results of our NQR, nonlinear response and specific heat investigations taken together with resonant X-ray scattering[30] enable us to form a complete picture of charge-stripe appearance, which we summarize in Fig. 5. Starting from high temperatures, nematic fluctuations appear and diverge at the well-defined nematic phase transition temperature $T_{EN}$. The nematic phase breaks orientational symmetry and it can be imagined as a stripe liquid with no translational symmetry breaking. This liquid then gradually freezes, with the first appearance of broken translational symmetry at $T_{CO}$ (as seen by X-ray scattering). However, the disorder present in the material strongly influences the stripe freezing, eliminating a sharp charge-stripe phase transition and making $T_{CO}$ a crossover temperature. Long-range stripe order is

never established, as evidenced by relatively small X-ray coherence lengths[31] and glassy dynamics observed in lanthanum NMR[14,39].

Our study clearly shows that charge stripes develop through a nematic phase. As LESCO was carefully chosen as a model material, this scenario could plausibly apply to cuprates in general, with some material-specific modifications. Perhaps most notably, the frequently studied YBa$_2$Cu$_3$O$_y$ also shows charge stripes in a dome-shaped region[1,3,6], but the formation of nematic order is more difficult to establish, as YBa$_2$Cu$_3$O$_y$ already has broken orientational symmetry due to Cu–O chains present in its structure. Furthermore, the recent observation of charge order[4] in the tetragonal HgBa$_2$CuO$_{4+\delta}$ (Hg-1201) might provide an opportunity to further test the phase diagram suggested here.

Finally, our investigation suggests that the high-temperature pseudogap is distinct from the well-defined nematic order we observe here. Both nematic and charge-stripe ordering tendencies are most prominent around 1/8-doping, whereas the pseudogap line has a different doping trend and occurs at significantly higher temperatures in the underdoped region. However, nematic effects have repeatedly been observed to appear together with the pseudogap[22–24], implying that a nematic component is present in the pseudogap state. Thus, within the scenario of Fig. 5, the pseudogap might be related to (or at least interact with) the high-temperature loop liquid, which is close to a nematic instability. More broadly, our finding of the unconventional nematic phase in LESCO provides a connection to other materials such as pnictides and demonstrates the power of using complementary experimental techniques—local probes such as NQR and unconventional transport measurements, such as nonlinear conductivity—in detecting strange metallic phases. This could be employed in answering fundamental questions in diverse correlated electronic systems, including heavy fermions and quantum magnets.

## Methods

**Samples.** For all experiments, high-quality single crystals of LESCO were used, with typical dimensions of a few mm$^3$. They were grown by Udo Ammerahl by the travelling solvent floating zone technique under 3 bar oxygen pressure in the group of A. Revcolevschi, Laboratoire de Chimie des Solides, Universite Paris-Sud. The $x$- and $y$-values were determined by energy-dispersive X-ray analysis and were consistent with the nominal values. The same crystals were used in different previous investigations, including NMR[32], X-ray[5,30,31] and thermal transport/Nernst effect[52] studies.

**NMR and NQR analysis.** In obtaining the wipeout fractions, spin–spin relaxation measurements were performed at the peak of the $^{63}$Cu A lines in LESCO. The entire lineshapes were not integrated, as it is known from previous work[13] that they do not appreciably change in the temperature range 20–200 K important for us. The Cu NQR lineshape remains the same all the way from $T_{LTT}$ down to very low temperatures, where it changes drastically due to spin order[13]. To additionally check this fact, we made an NMR measurement of the $^{63}$Cu quadrupolar satellite of LESCO-1/8 in an in-plane magnetic field of 11T (Supplementary Fig. 3). The spectrum (and thus NQR parameters and linewidth) changes very little from 60 to 110 K, justifying our assumption of temperature-independent NMR/NQR lineshapes.

Spin–spin relaxation was compensated, for using purely exponential fits, as we did not notice any significant departures from exponential relaxation in all samples in the relevant temperature range. Supplementary Fig. 1 shows selected NQR spin–spin relaxation curves in three LESCO samples, whereas the NMR measurements on the line shown in Supplementary Fig. 3 is in Supplementary Fig. 2. The signal-to-noise ratio in the NQR measurements is relatively poor due to skin depth effects and comparatively low frequency of ~36 MHz. NMR results (at ~140 MHz) have a much better sensitivity and hence smaller error bars in the wipeout determination. Within our experimental accuracy, we find no Gaussian components in the spin–spin relaxation in either NMR or NQR. Despite the low signal-to-noise in LESCO $x=0.11$ and $x=0.125$, the wipeout below $T_{EN}$ is large enough to be reliably resolved. The NMR measurement confirms its existence with much better precision. Wipeout in Cu NMR for $x=0.125$ and NQR for $x=0.15$ was measured up to 170 K, to check whether the LTT transition influences the wipeout fraction. No such influence was found, in agreement with all previous investigations. The structural transition is close to 130 K in the entire investigated

doping range[31,40] and thus cannot influence the wipeout step. Owing to high conductivity, the skin depth in LESCO is below 10 μm (at the NQR frequency of ~ 36 MHz) in all investigated samples, making the NQR signal small. We stress that the changes in conductivity at MHz frequencies are of the order of 10% in our temperature range, too small for the change in skin depth to account for the modified NQR signal.

**Nonlinear conductivity.** The nonlinear conductivity was determined using a two coil radiofrequency (RF) inductive method. The setup is similar to the well-known microwave conductivity experiment[53,54] but uses RF excitation fields. The sample was mounted on a sapphire rod in a vacuum tube, with two coils wound around the tube and immersed in liquid helium. RF pulses were supplied to one of the coils using an NMR spectrometer, whereas the third harmonic signal was detected on the other coil. The setup and measurement procedure are described in detail in ref. 41. In experiments on LESCO, we have used RF pulses at an excitation frequency of 18.8 MHz, with pulse lengths of 50 μs, repetition time 8 ms and maximum pulse power ~ 100 W. Measurements were performed with the power varying from 10 to 100% full power and the low power points adjusted to match at all temperatures to avoid baseline drift. Error bars reflect the accuracy of this adjustment.

**Heat capacity.** The specific heat of LESCO was measured using a sensitive differential thermal analysis (DTA) setup[55]. The principle of operation is simple: the sample of interest and a reference are mounted on resistance thermometers, which are connected to a large mass platform with weak heat links. If the temperature of the platform is swept linearly in time, the temperatures of the sample and reference will lag behind the platform temperature by an amount proportional to their respective specific heats. If the temperature difference between sample and reference is detected, a sensitive differential specific heat measurement can be obtained. This kind of DTA is especially suited for small and sharp features in the specific heat—a conventional method such as pulse calorimetry does not have the necessary resolution to detect the small features in LESCO-specific heat occurring at $T_{EN}$. However, exact absolute specific heat values are hard to extract from the DTA measurement, which is why we only focus on sharp features in the differential heat capacity. The resolution of the method is limited by the resolution of temperature difference determination, which can easily be made very accurate.

In our setup (Supplementary Fig. 6) we use a home-made sample probe inserted either in a variable temperature inset of a superconducting magnet or a liquid helium/nitrogen bath. Platinum resistors are used as temperature sensors in a bridge configuration, with lock-in amplifier detection of the unbalance voltage. The sample is mounted on the small-profile platinum resistor with a small amount of Apiezon N grease to increase thermal contact. Samples, thermometers and the heating/cooling platform are held in high vacuum and the temperature of the platform is separately monitored and regulated with a heater and Cernox temperature sensor.

The setup was not calibrated to measure absolute values of $C_p$ (hence, the arbitrary units in Fig. 4), as we are only interested in temperature-dependent features of $C_p$ and not its absolute value. The measurement always includes a smooth background due to sample-specific heat, mismatch between the two platinum chips and possible other imperfections; for us it was important to observe sharp features at the structural transition $T_{LTT}$ and the hidden ordering temperature $T_{EN}$; thus, we modelled the background as a simple fourth-order polynomial and subtracted it. The peak at $T_{LTT}$ (which agrees precisely with values from X-ray scattering) is a convenient demonstration of the sensitivity of the method and at the same time clearly shows that the ordering at $T_{EN}$ is distinct from the structural effects in the vicinity of $T_{LTT}$. We note that the charge ordering at $T_{CO}$ is not a true phase transition in 214 cuprates due to disorder and should not have a sharp signature in the specific heat. It would most probably show up as a low and broad feature below $T_{CO}$, virtually undetectable with our current setup.

The two $\Delta C_p$ measurements shown in Fig. 4 were obtained in different conditions to check the repeatability and resolution of the experiment. The data in the main graph is from a measurement in a liquid nitrogen bath, with open-loop temperature sweep control (that is, a linear sweep of the heater power)—this minimizes small temperature oscillations, which influence the measurement. The typical heating rate in that experiment is ~ 1 K min$^{-1}$. The data in the inset are from a slow measurement with the entire probe in a variable temperature inset (in helium gas). The system was heated at a rate of ~ 0.1 K min$^{-1}$, with lock-in filter times of 100 s to increase signal-to-noise. The slow measurement is better suited to determine the sharpness of the transition at $T_{EN}$, but is also much more sensitive to unwanted temperature oscillations and thus more difficult. It is also clear that no sharp features occur at or near the charge-stripe ordering temperature $T_{CO}$, as should be expected due to disorder. An additional measurement of the specific heat of the sample with $x = 0.15$ was performed in a nitrogen bath in a range close to $T_{EN}$. The results and comparison to the $x = 0.125$ measurement (obtained in similar conditions) are shown in Supplementary Fig. 5.

**Data availability.** The data supporting the findings of this study are available within the article and its Supplementary Information file.

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

## Acknowledgements

We thank B. Büchner for providing the LESCO single crystals, N. Bonačić for assistance with the specific heat measurements and M. Grbić, A. Dulčić and M.-H. Julien for comments. D.P., M.V. and M.P. acknowledge funding by the Croatian Science Foundation under grant number IP-11-2013-2729 and S.-H.B. acknowledges support by the DFG Research Grant Number BA 4927/1-1.

## Author contributions

D.P. conceived the study, performed NQR, nonlinear conductivity and specific heat measurements, analysed data and wrote the paper. M.V. designed and constructed the nonlinear conductivity setup and performed measurements. H.-J.G. selected samples and participated in study design and data analysis. S.-H.B. was involved in designing the study and results analysis. M.P. performed NQR measurements and supervised the project. All authors took part in discussing results and editing the manuscript.

## Additional information

**Competing financial interests:** The authors declare no competing financial interests.

