## [Peer review file · Nature Communications]

Reviewers' Comments:

Reviewer #1 (Remarks to the Author)

Motivated by experimental observations of CDW correlations and theoretical developments of a closely related state in cuprates that preserves translational symmetry, Pelc *et al* investigate the process of charge stripe formation and the possible existence of an intermediate electronic nematic phase in LESCO via three different experimental techniques: NQR, non-linear conductivity, and specific heat. The main results appear to be:

1) For NQR, LESCO samples of three different dopings all exhibit a drop in ^{63}Cu NQR signal intensity ("wipeout") at temperatures higher than T_{CO} where strong CDW fluctuations set in. Assuming the wipeout effect is due to a phase transition related to charge order, the authors conclude that the transition temperatures T_{EN} closely trace T_{CO} as doping varies.

2) The authors then measure the (third-order) non-linear conductivity of the system, which exhibits a sharp peak at the previously-determined T_{EN} . The technique was benchmarked on iron-based superconductor in a previous paper (Ref [38] in the main text). The authors also notice that the linear conductivity does not show any prominent feature at T_{EN} .

3) The authors also perform specific heat measurement and detect a peak in the differential heat capacity at T_{EN} . The technique is benchmarked by the LTO-LTT transition. Interestingly, no sharp features are present at T_{CO} , suggesting that the charge stripe transition in the material is significantly rounded by disorder.

The results are novel and of great interest to high- T_c community, both experimentally and theoretically, especially the specific heat measurement which as far as I know is the first direct thermodynamic evidence of a new phase transition above the charge ordering temperature T_{CO} in La-based cuprates. The presentation of the results is clear. However, I would not recommend the current version of the manuscript to be published in Nature Communications, because more data is needed to make the results complete (see Question 1 below) and the conclusion convincing (see Question 2 below), and the implications of some of the results are not clearly discussed (see Questions 3, 4, 5 below). I hope the authors can address the following questions:

1) I noticed that in Fig. 1 all three NQR signals terminate below the LTT transition. Is it possible to include data above the LTO-LTT transition? It would be helpful for us to better understand the wipeout effect by looking at the NQR response across that transition.

2) Based on the transition temperatures at different dopings, the authors conclude that there is a close relationship between stripe order and the intermediate phase. However, the authors have adopted slightly different criteria to determine T_{CO} and T_{EN} in Fig. 1: T_{EN} seems to be defined as the temperature where the the drop in signal intensity reaches half of the plateau height, whereas the T_{CO} coincides with where the intensity {it starts} to drop. Although it does not make too much a difference, it would still be better to have a consistent definition of transition temperatures from NQR signals. More importantly, since wipeout effect is not yet thoroughly understood, it would be better to use other methods to pin down the nematic transition temperatures instead using solely NQR results. In particular, could the specific heat measurement be done on the samples with doping concentrations 0.11 and 0.15?

3) Last sentence in the abstract: "...show that the nematic phase is unrelated to high-temperature pseudogap physics,...", as well as Page 2, second paragraph, last sentence: "...implying that the high-temperature pseudogap physics is unrelated to nematic / charge stripe occurrence" -- is there direct evidence from the experiments for this claim? Even if this is true for LESCO, in other materials of cuprate family (e.g. YBCO) it has long been proposed (based on transport measurements) that the nematic transition/crossover has an onset temperature that increases with decreasing doping, and traces the pseudogap temperature(s) observed in ARPES, NMR, Kerr, etc., suggesting that nematicity is one facet of the pseudogap physics. Therefore, the authors' claim may need certain refinement.

4) The third-order non-linear conductivity should be a rank-4 tensor. It would be better if the authors provide more information about which component(s) of the tensor is(are) being measured.

5) It should be noticed that in the LTT phase, there is effectively already a nematic order within each Cu-O plane, but it rotates from plane to plane so there is no global nematicity. Although the authors have provided evidences of the existence of a phase transition (at least a strong crossover) at T_{EN} , none of the measurements shows directly that global C_4 rotation symmetry is broken at T_{EN} . Moreover, it is not clear from the present study about the exact pattern of the nematic order (unless the non-linear conductivity gives certain symmetry-related information, which goes back to Question 4 above). Is there any crucial information from the experiments that I am missing?

Reviewer #2 (Remarks to the Author)

The authors combined NQR, conductivity, and heat capacity techniques to study the Sr/Eu co-doped LSCO, and claimed that they observed a new "charge nematic phase". I do not recommend the paper for publication based on the three major reasons appended below:

1, The topic has been discussed extensively, as partly reflected in their references. This manuscript does not really add much new information, and is therefore lack of novelty. If one still wants to address this issue, I would prefer using some more direct technique, such as x-ray diffraction (Refs. 1,3,28).

2, Their data cannot support the existence of the "charge nematic phase". If one does not consider the inhomogeneity of the sample co-doped with Sr and Eu (Ref. 34), and assumes the sample is homogeneous, excluding all possible extrinsic effects (for instance, caused by Eu), their data is still very shaky in supporting their claims. For instance, from their NQR data as shown in Fig.1, I do not see how the authors get the transition temperatures for the "charge nematic phase". The features are too weak and all of the data points in the region of interests are within the uncertainties.

3, The manuscript overall were poorly written, and many statements will need to be rewritten. Take several as examples, in the abstract, it is stated that "it is unclear how it emerges from the parent metallic state." Do the authors consider the parent compound La_2CuO_4 metallic?? Again in the abstract, "show that the nematic phase is unrelated to high-temperature pseudogap physics". Isn't it too bold to make this kind of arguments based on the limited evidences? The title, "Tc"; figures, "temperature (K)"...

Reviewer #3 (Remarks to the Author)

The authors present a thorough study of La-Eu-Sr-Cu-O crystals with three different Sr concentrations by means of NQR (NMR), non-linear ac conductivity and differential calorimetry. The resulting phase diagram suggests existence of an additional, T_{EN} , intermediate phase transition temperature that approximately follows the charge order, but appears not to be correlated with a pseudogap temperature. The authors speculate existence of the charge nematic state for the LESCO materials and extend this speculation to other cuprates.

The manuscript is well written with clear explanation (mainly in SI) of the experimental techniques used.

Although existence of charge nematic state is somewhat speculative, in my opinion this conjecture might be interesting for the community and altogether the manuscript is suitable for publication in Nature Communications.

The only criticism I have is that the authors provide no information about the samples used: how were they grown and by which group? how the x and y values were determined? how the quality of the samples was assessed? what other measurements were performed on these samples?

Reviewer #1:

We acknowledge the constructive criticism of Reviewer #1, and have performed additional measurements in order to address their concerns. Answers to specific points:

1) we have measured the signal intensity for two samples ($x = 0.15$ in NQR, and $x = 0.125$ in NMR) up to ~ 160 K, to show that no appreciable effects are visible near the LTT transition temperature. This is in agreement with all previous wipeout investigations we are aware of (with the exception of LBCO-1/8, where wipeout is caused by charge order since T_{LTT} and T_{CO} almost coincide – e.g. Ref. 36 in the manuscript).

2) the additional measurement of specific heat for the sample with $x = 0.15$ was performed, and transition temperatures obtained from specific heat are now added to the phase diagram in Fig. 2. Unfortunately, the sample with $x = 0.11$ was lost during one of the NQR experiments, so no further measurements on it were possible. Yet since T_{EN} values from NQR and specific heat agree well for the other two samples, it is reasonable to assume that $x = 0.11$ is no exception (especially since the wipeout step is clearest for that doping). We note that T_{CO} in Figs. 1 and 2 is not from NQR, but from resonant X-ray (Refs. 30 and 31). These temperatures agree with the onset of further wipeout, which is expected since T_{CO} is not a phase transition, but rather an onset temperature.

3) it is clear from our results that T_{EN} is significantly below the pseudogap temperature T^* , and has a different doping trend in the investigated range. Thus one cannot identify T^* as a true nematic ordering temperature in LESCO (which we take as a model stripe-ordered cuprate). However, we agree that nematicity has been observed in the context of pseudogap physics, and believe this might be due to a connection to the high-temperature stripe liquid which is close to a nematic instability (Fig. 5 in the manuscript). In materials such as YBCO, the weak structural orthorhombicity could then induce electronic orientational symmetry breaking. Testing this idea would be an interesting continuation of the present work. The manuscript has been modified (in the abstract, introduction, and discussion sections) in order to accommodate the above considerations.

4) & 5) the tensor nature of third harmonic response is now discussed in detail in the Supplementary Information (along with a modified section after eq. (1) in the main text). We also include a heuristic derivation of the relationship of the tensor components with nematic fluctuations. Namely, third harmonic response is not symmetry-constrained - its appearance is allowed by symmetry in both nematic and isotropic (or in this case C2 and C4) situations. Thus the signal is not proportional to the nematic order parameter, as would be the case with a symmetry-constrained quantity. Instead, third order response components are proportional to four-point correlation functions, which are a direct measure of nematic fluctuations. Thus σ_3 is the nematic analogue of e.g. magnetic susceptibility in a ferromagnet: it is sensitive to fluctuations of the order parameter, and not the symmetry breaking itself. This is now discussed in the Supplementary Information.

Reviewer #2:

1) we agree that nematicity in strongly correlated systems is an area of intense investigation, but believe that our work adds significant novelty to the discussion in cuprates. Namely, we identify that (i) the nematic phase is well defined thermodynamically, in contrast to charge stripe order, and (ii) using nonlinear response, show a divergence of nematic fluctuations, which to our knowledge was never before observed in the cuprates. Using X-ray scattering in studies of nematic order is difficult for two reasons: a nematic does not break translational symmetry, and X-ray scattering probes extremely short timescales, making it impossible to separate fluctuations from true long-range order. This is why we have opted to combine a local technique (NQR) with bulk probes (nonlinear response and specific heat) to obtain an overarching picture of charge nematic/stripe formation, which would not have been possible otherwise.

2) in our work the appearance of a new phase at temperatures above the charge order onset is consistently observed in three different experimental techniques. In particular, the NQR wipeout plateau is significantly larger than the error ranges in all three samples (and in Cu NMR on the sample with $x = 0.125$). The identification of the new phase as a nematic rests on two experimental facts: a divergence of the third order response, which is a measure of nematic fluctuations (the connection is now described in more detail – see also reply to Reviewer #1 questions 4 & 5); and the insensitivity of the phase transition on disorder, which has been predicted theoretically for a charge nematic. Although we cannot make a definitive identification, the experimental data strongly suggest that the intermediate phase is a nematic.

3) the remarks have been taken into account in the revised text. 'Parent metallic state' did not refer to the parent compound (which is of course an insulator), but to the high-temperature state from which the charge order appears upon cooling. As for the connection with pseudogap, our data clearly show that T_{EN} is not T^* in LESCO, but we agree that the question of pseudogap and nematicity is more complex than what was written in the first version of the manuscript (see also the answer to Reviewer #1 question 3). This is now amended through a significant change in the discussion, and modifications in the abstract and introductory paragraphs. The minor typographical/style errors pointed out by the Reviewer have been corrected.

Reviewer #3:

We agree that the samples were not described in detail in the previous version – they are indeed well characterized and were used in different previous experiments. This is now described in the methods section, and references to earlier work have been added.

Reviewer #1 (Remarks to the Author)

Questions (1), (2) and (3) have been addressed satisfactorily by the authors. About Questions (4) and (5): The heuristic derivation added by the authors of the relationship between σ_3 and nematic fluctuations indeed provides a preliminary way to understand the conductivity measurement. However, since the conclusion that the transition is nematic relies solely on the conductivity measurement, I strongly suggest adding a more rigorous derivation/definition that unambiguously demonstrates the direct relation between non-linear conductivity and nematic fluctuations. This could be achieved via either presenting the derivation explicitly or providing page reference of Ref [10] in Supplemental Materials (Ref [47] in Main Text). Note that, although it is generally true that non-linear susceptibility is proportional to four-point correlation function (as discussed in Ref [9] of Supplemental Materials), Ref [9] does not seem to include an explicit relation between non-linear conductivity and two-point nematic correlation function, which is relevant to the authors' study. I'd also like to clarify that certain components of the third harmonic response are in fact symmetry-constrained, but it might be difficult to measure a single component of σ_3 experimentally. For example, $\sigma_{3,xxxy}$ should vanish under the presence of both C4 and diagonal mirror ($x \rightarrow y, y \rightarrow x$) symmetries, but may acquire a nonzero value when C4 is broken to C2.

Reviewer #2 (Remarks to the Author)

My questions and concerns have been addressed in the rebuttal letter, but I am still not quite convinced with the results, especially the NQR data, as well as the conclusions derived from. Anyway, I'll leave it to the editor to decide whether to accept the manuscript or not. If it is to be published, I suggest changing the order of the presentation. Look to me that it may be appropriate to show the heat capacity data first (e.g., Fig.3).

Reviewer #3 (Remarks to the Author)

In my opinion the revised version contains reasonable responses for the referees' comments and is suitable for publication in Nature Comm.

Reviewer #1:

We have further expanded the discussion of nonlinear response in relation to nematic fluctuations, with two points worth emphasizing:

(i) a derivation of the relationship between the components of σ_3 and the nematic susceptibility χ_{EN} for the case of tetragonal symmetry is now added. While there are several simplifying assumptions (such as neglecting interaction terms and working in the static limit), we believe that the derivation provides the necessary link suggested by the Referee. We note that a similar calculation has been performed for nematic fluctuations in liquids, e.g. in Ref. 13 of the revised Supplement (such a derivation to our knowledge is not present in Ref. 14, so a page reference is not provided).

(ii) it is true that Ref. 8 in the revised Supplement does not include an explicit relation between nematic (quadrupolar) fluctuations and third order response – the reason is that Ref. 8 treats the most general case of a random material. We therefore include new references 9 and 10 in the Supplement (and Ref. 46 in the main paper) where the case of quadrupolar correlations is treated explicitly. Especially Ref. 9/46 is directly analogous to our case.

As the Referee notes, some components of the σ_3 tensor are symmetry-constrained, but are impossible to separate from other contributions in our experiment. This is now commented in the Supplement.

Reviewer #2:

Prompted by the Referee’s suggestion of changing the order of the presentation, we have considered rearranging the paper to place the specific heat and/or nonlinear conductivity data before NQR. The reason for presenting NQR first is that it gives an indication for the existence of the intermediate phase, but offers no direct clues of its nature; nonlinear conductivity and specific heat are then used as additional follow-up probes to provide a complete picture. After considering alternative options, we still feel that this is the arrangement which is easiest to follow. Yet if the Editor thinks that a different order is more suitable, we are ready to reconsider the structure of the manuscript.

Reviewers' Comments:

Reviewer #1 (Remarks to the Author)

The concerns from my previous review have been addressed by the authors. A minor point: I noticed that in Supplementary Figure 5 the authors have added the specific heat data of doping=0.15. It would be better if the non-linear conductivity data (as a function of temperature) of doping=0.15 is plot together with doping=0.125 as well. Other than that, I recommend the manuscript being accepted.

Reviewer #1 (Remarks to the Author):

The concerns from my previous review have been addressed by the authors. A minor point: I noticed that in Supplementary Figure 5 the authors have added the specific heat data of doping=0.15. It would be better if the non-linear conductivity data (as a function of temperature) of doping=0.15 is plot together with doping=0.125 as well. Other than that, I recommend the manuscript being accepted.

Answer:

We thank the Reviewer for the positive evaluation of our previous responses. As a minor point the Reviewer suggests that 'it would be better if the non-linear conductivity data (as a function of temperature) of doping=0.15 is plot together with doping=0.125 as well.' Unfortunately, the data mentioned by the Reviewer do not exist - in the manuscript on page 5 we state that nonlinear conductivity was only measured for doping 0.125. This doping was taken as representative based on the phase diagram from NQR and previous X-ray work, and we believe that the presented results leave no doubts about our main conclusions. The Reviewer suggests that it would be better (but not compulsory) to show this data. While it would be possible to measure nonlinear conductivity also for doping 0.15, this experiment is rather involved and would take a long time, and we are not convinced that the additional measurements would bring a new quality to the manuscript. In order to avoid postponement of the manuscript publication, we would prefer not to wait for this measurement for the present paper.